
# Technical note: A revised incoming neutron intensity correction factor for soil moisture monitoring using cosmic-ray neutron sensors

Magdalena Szczykulska[1], David Boorman[1], James Blake[1], Jonathan G. Evans[1]

[1]UK Centre for Ecology & Hydrology, Wallingford, Oxfordshire, OX10 8BB, UK

*Correspondence to*: Magdalena Szczykulska (magszc@ceh.ac.uk)

**Abstract.** The cosmic-ray neutron sensor method of soil moisture measurement is now widely used and is fundamental to the COSMOS-UK soil moisture monitoring network. The method is based on a relationship between a measured flux of neutrons and soil moisture, and requires the neutron count to be adjusted for time variations of atmospheric pressure, humidity and the incoming flux of cosmic-ray neutrons. This note describes an empirical approach to the development of a revised correction

factor for the last of these. Using the revised correction factor makes a significant difference to the derived soil moisture at wetter sites. This has implications for quantifying the soil moisture regime at these sites and management decisions that depend on a proper understanding of soil moisture dynamics, such as flood management and the release of greenhouse gases.

## 1 Introduction

Soil moisture plays a pivotal role in a range of land surface processes including the transport of energy and matter via

evapotranspiration, drainage, run-off infiltration and plant photosynthesis (Dorigo et al., 2017; Brocca et al., 2017). Properly measuring soil moisture enables a better understanding of these processes and can inform management decisions. As well as benefiting agriculture and drought management, there are many other, perhaps less obvious, ways in which soil moisture data are beneficial. For example, the wetness of peat soils is a significant control on the emission of greenhouse gasses (Evans et al., 2021), and when these soils dry there is the potential for wildfires (Sungmin et al., 2020). In addition, the rewilding of

upland areas is seen as a way of mitigating flood risk by providing additional storage capacity (Keesstra et al., 2018). Understanding the soil moisture regime is important in understanding and quantifying the potential benefits of such schemes, and knowing the soil moisture status in real-time can aid flood forecasting (Silvestro et al., 2019).

The cosmic-ray neuron sensor (CRNS) method is now widely used to measure soil moisture (Zreda et al., 2008; Kohli et al., 2015; Andreasen et al., 2017; Cooper et al., 2021; Upadhyaya et al., 2021). In outline, the method uses a cosmic-ray neutron

sensor situated above the land surface to provide a raw count of epithermal neutrons at the soil moisture site (SMS). This count rate is then corrected for factors which influence the flux of neutrons arriving at the Earth's surface, typically time variations in the background neutron intensity (incoming flux of cosmic-ray neutrons), atmospheric pressure and humidity.





Soil moisture, usually expressed in the form of volumetric water content (VWC), is then derived from the corrected count rate, $N_{corr}$, using a relationship of the form:

$$VWC = \left( \frac{0.0808}{\frac{N_{corr}}{N_0}-0.372} - 0.115 - (\tau + SOC) \right) \frac{\rho_{bd}}{\rho_w} \qquad (1)$$

where $\tau$ is the lattice and bound water, $SOC$ is soil organic carbon, $\rho_{bd}$ is the bulk density of the soil at the SMS, $\rho_w$ is the water density equal to 1 g cm$^{-3}$ and $N_0$ are the dry corrected counts on the day of calibration. Equation (1) is derived from numerical neutron modelling (Desilets et al., 2010) alongside independent field measurements of soil moisture for calibration to determine $\tau$, $SOC$, $\rho_{bd}$ and $N_0$ (Evans et al., 2016; Franz et al., 2013; Zreda et al., 2012). Note that in other studies a biomass

correction is introduced (Baatz et al., 2015).

The simplest representation of the factors used to calculate the corrected count rate, $N_{corr}$, is:

$$N_{corr} = N_{raw} \frac{(F_q . F_p . F_o)}{F_i} \qquad (2)$$

where $N_{raw}$ is the uncorrected count rate, $F_q$ is the humidity correction factor, $F_p$ is the atmospheric pressure correction factor, $F_i$ is the background neutron intensity correction factor and $F_o$ is a placeholder introduced by (Zreda et al., 2012) to allow for

other future correction factors, a useful reminder of the unknowns associated with the CRNS method, although in practice $F_o$ is taken as one.

In its simplest form the background neutron intensity correction factor, as given by (Zreda et al., 2012), is:

$$F_i = \frac{I}{I_{ref}} \qquad (3)$$

where $I$ is the count rate of incoming cosmic-ray neutrons at a high resolution reference neutron monitoring site (RNMS) at

the time of interest and $I_{ref}$ is the incoming count rate at an arbitrary time (e.g. network start date or the date of the in situ CRNS calibration; note that $I$ and $I_{ref}$ have already been efficiency and pressure corrected at source). $F_i$ therefore represents the normalised background neutron intensity as recorded at the RNMS.

(Hawdon et al., 2014; Schrön, 2017) reference an alternative equation for the neutron intensity correction factor:

$$F_i = \gamma \left( \frac{I}{I_{ref}} - 1 \right) + 1 \qquad (4)$$

in which $\gamma$ is a constant for a specific SMS. (Hawdon et al., 2014) suggest that $\gamma$ may be estimated from the cut-off rigidities of the SMS and RNMS, but in practice it is usually set to 1 and Eq. (4) reduces to Eq. (3).

COSMOS-UK (Cooper et al., 2021) accesses the RNMS data from the Neutron Monitoring Database (nmdb.eu) by default uses data from Jungfraujoch (JUNG) provided by the Physikalisches Institut, University of Bern, Switzerland. Daily corrected



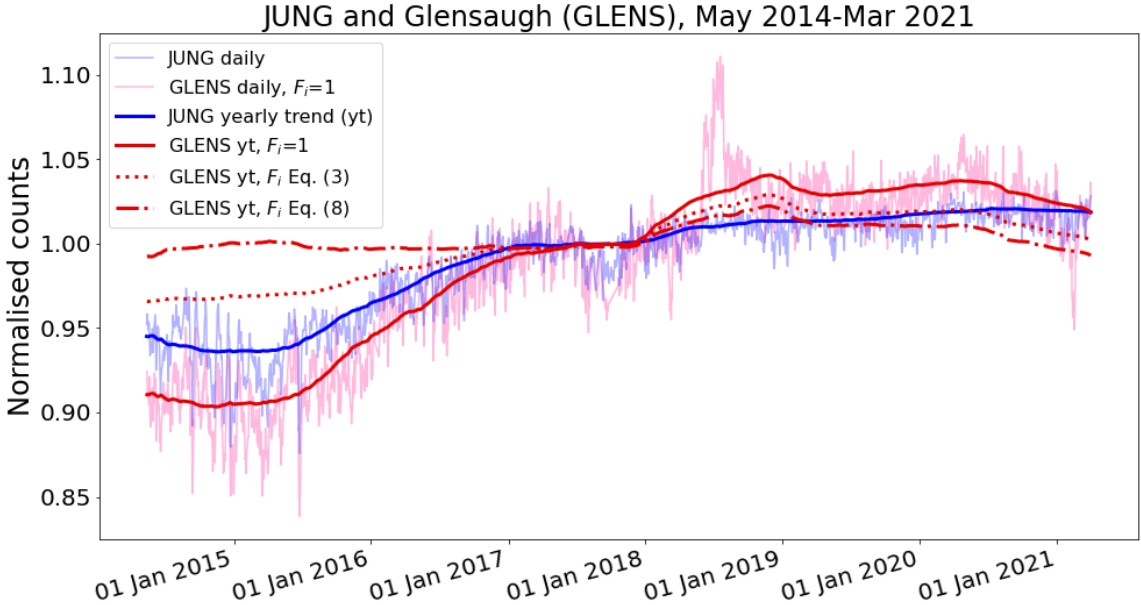

**Figure 1: Normalised counts from the JUNG RNMS daily data (pale blue) and 365 day running mean (blue); and from the GLENS SMS daily data (pink) and 365 day running means (red) using three different correction factors. The daily counts and their running means are normalised to a median of values for the given time period.**

neutron count data from JUNG are shown in Fig. 1 alongside corrected CRNS counts at Glensaugh SMS. The CRNS counts

in the figure are corrected using three types of neutron intensity corrections: $F_i = 1$ (no intensity correction), the standard correction given in Eq. (3) and the empirical correction proposed in this work given in Eq. (8). While the standard intensity correction in Eq. (3) partially removes the JUNG trend, some of it still remains.

## 2 Method

### 2.1 An empirical approach to a revised correction factor

Equation (3) implies that the correction is dependent on the neutron flux as measured by the RNMS, with the assumption that the RNMS adequately represents the relevant incoming neutron flux at the SMS. However, the correction should also account for the differences in detector and location characteristics between the SMS and the RNMS, and the introduction of the $\gamma$ constant in Eq. (4) was intended to account for at least some of these differences.

One approach to quantifying these differences would be to use a neutron transport model based look-up table, to estimate

neutron flux scaling factors between different locations (e.g. http://seutest.com/cgi-bin/FluxCalculator.cgi). The look-up tables are essentially based on cut-off rigidity and altitude, and if the cut-off rigidity is treated as being constant in time, then the





scale factors applied to $N_{corr}$ and $N_0$ in Eq. (1) cancel out. A constant scale factor applied to counts does not have an impact on VWC due to this cancelation.

An alternative approach explored in this technical note is to derive an empirical correction factor by comparing the count rates at the SMS, $N$, and at the RNMS, $I$, which is expected to yield more accurate results as *in situ* measurements of neutron fluxes at both sites (as recorded by the different detectors) are directly compared, albeit with the necessity to reduce the effect of soil moisture variations on the SMS.

Firstly, $N_{raw}$ is corrected for humidity and pressure, to provide $N$, then both $N$ and $I$ are standardised. A wide range of standardisations is possible; the form adopted here is

$$N_n = \frac{N}{N_m} - 1 \tag{5}$$

where $N_n$ is the normalised count rate, $N_m$ is the median count rate over the period of operation of the SMS. The median provides a very stable central estimate of the count rate; subtracting one means that the normalised count rate will vary about zero. The RNMS count rate is normalised in the same way to give $I_n$ using the median count rate over the same period as for the SMS.

Normalised count rates have been derived in this way for data from the COSMOS-UK network, and a plot of $N_n$ against $I_n$ is presented in Fig. 2a for the Glensaugh SMS. This shows a very convincing linear relationship, even though the SMS count rate is expected to be greatly influenced by soil moisture whereas this is not expected to be an effect at the RNMS. The Glensaugh site is one of the wettest sites in the COSMOS-UK network and in the period of operation from 2015, only in the summer of 2018 there was significant drying of the soil. A lower quartile regression, the choice of which is explained later in this section, between $N_n$ and $I_n$ gives a gradient, $G$, not of 1, but 1.5.

A similar plot from a SMS that does regularly dry in the summer months (Bunny Park) is shown in Fig. 2b, and the relationship between the count rates shows more scatter. However, by differentiating the data by season, the wetter winter months are located at the bottom of the cloud of points and show significantly less scatter. As with data from Glensaugh, it is expected that soil moisture is relatively constant during these months. In contrast to this, soil moisture in the summer months is extremely variable.

There are various methods available to abstract a gradient from these data e.g. ordinary least squares linear regression with and without the intercept, quantile regression, fitting to winter data only, fitting to 'wet' data only.

In practice, all of these yield rather similar estimates of $G$ at all COSMOS-UK sites, and if the regression includes an intercept term this is small, and not significantly different from zero. Results presented here are from quantile regression fitting to the first quartile (quantile = 0.25). This was seen as an objective way of fitting to the lower part of the cloud of data points. At Bunny Park this gives $G$ of 1.2.





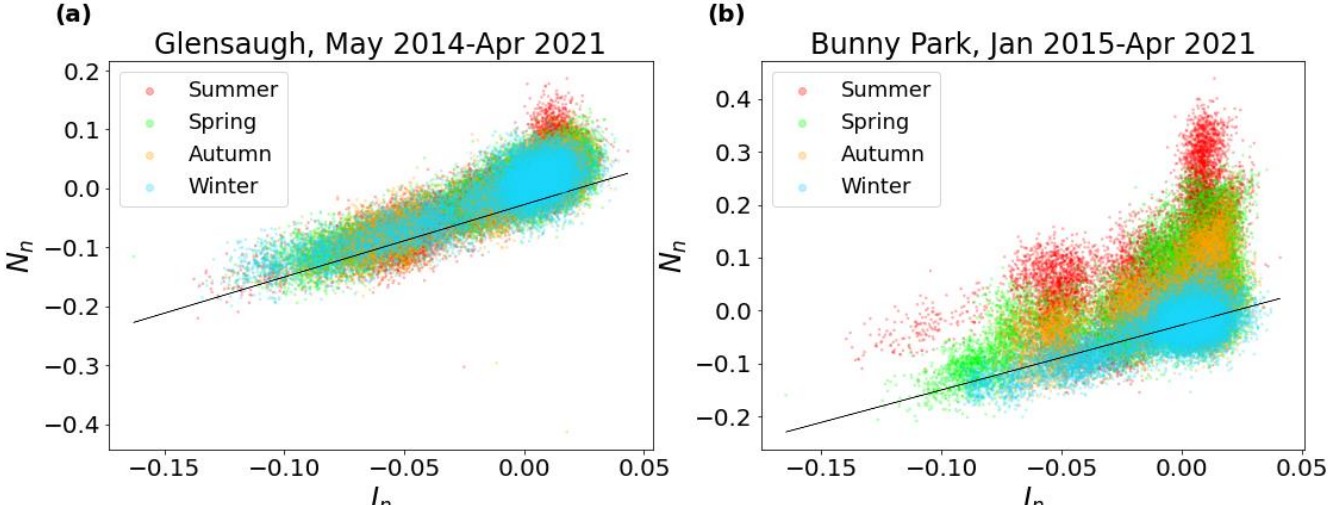

**Figure 2:** $N_n$ **plotted against** $I_n$ **for two COSMOS-UK SMSs, Glensaugh (a) and Bunny Park (b). The points are coloured by season, cyan, green, red and orange representing winter, spring, summer and autumn respectively. The plotted lines are from quantile regression fitting to the first quartile.**

If the intercept term from the regression is ignored, then $G$ scales $I_n$ to provide an estimate of $N_n$, $\widehat{N}_n$, biased towards wetter periods where soil moisture is more stable:

$$\widehat{N}_n = G I_n. \tag{6}$$

Using Eq. (5), $\widehat{N}_n$ can be rewritten as $\widehat{N}_n = \left(\frac{N'}{N_m} - 1\right)$ where $N$ is replaced with $N'$, CRNS counts at relatively constant soil moisture at the SMS. Substituting for $\widehat{N}_n$ and $I_n$ in Eq. (6) gives:.

$$\frac{N'}{N_m} = G\left(\frac{I}{I_m} - 1\right) + 1 \tag{7}$$

Time variations in $N'$ are more representative of the time variations in the incoming neutron counts at the SMS when compared to $I$. Equation (7) is therefore the desired alternative to Eq. (3) and forms the revised neutron intensity correction factor at the SMS. It is worth noting that the normalisation here is performed with respect to the median of counts for the available time period rather than a specific date. COSMOS-UK SM data use $I_{ref}$, reference JUNG counts on the day of calibration, rather than $I_m$ in the right hand side of Eq. (7). This is because of the way the correction factor is currently implemented in the data system and in practice the difference between $I_m$ and $I_{ref}$ is insignificant. The revised correction factor therefore is:

$$F'_i = G\left(\frac{I}{I_{ref}} - 1\right) + 1. \tag{8}$$

It is interesting to note that Eq. (8) is identical in form to Eq. (4) and therefore $G$ can be considered to be an empirical estimate of $\gamma$.





## 2.2 Application

The new scaling factor $F_i'$ can now be used instead of Eq. (3) to revise $N_{corr}$ in Eq. (2). This is used firstly to calculate the parameter $N_0$ in Eq. (1) based on data obtained during field calibration, and then to derive a revised VWC time series.

The derived monthly mean soil moisture plots, obtained using intensity corrections $F_i'$ (Eq. (6)) and $F_i$ (Eq. (3)), are shown in
Fig. 3a and b for Glensaugh and Bunny Park respectively. Monthly data are used here to remove the very short term variations in VWC and make changes in VWC clearer. At Bunny Park, the maximum difference due to the revised intensity correction in monthly mean VWC is around 3%. At Glensaugh however the change is very much greater. The downward trend in VWC prior to 2018 is removed and there is better consistency with other data recorded at the same site.

One example of this better consistency comes from comparing the derived VWC data with the two point soil moisture probes
(time domain transmissometry, TDT, sensors) installed alongside the CRNS (Fig. 4). There are well known problems with TDT sensors, most notably the period at the start of the record while the sensors 'bed-in' and make a good connection with the surrounding soil following installation, and the impact of local changes in the soil matrix through cracking, root growth and burrowing animals. Such local changes are a possible explanation of the jump in VWC recorded in late 2020. Although these uncertainties concerning the accuracy of these data exist, it is clear that neither of these sensors shows a long-term drying
trend.

The extreme drying event of 2018 recorded by all three sensors has possibly caused a change in the soil moisture regime that warrants further monitoring and investigation, but is beyond the scope of this paper.

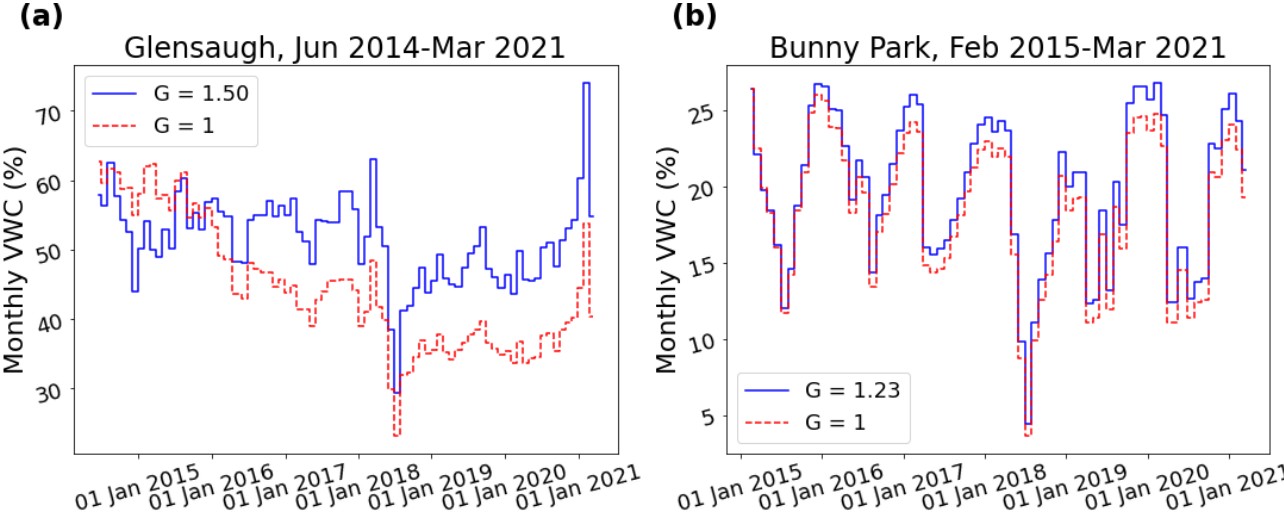

**Figure 3: Derived VWC from two SMSs, Glensaugh (a) and Bunny Park (b) using *G* of 1 (i.e. Eq. (3)) in red and *G* derived from**
**quantile regression fitting to the first quartile (i.e. Eq. (8)) in blue.**



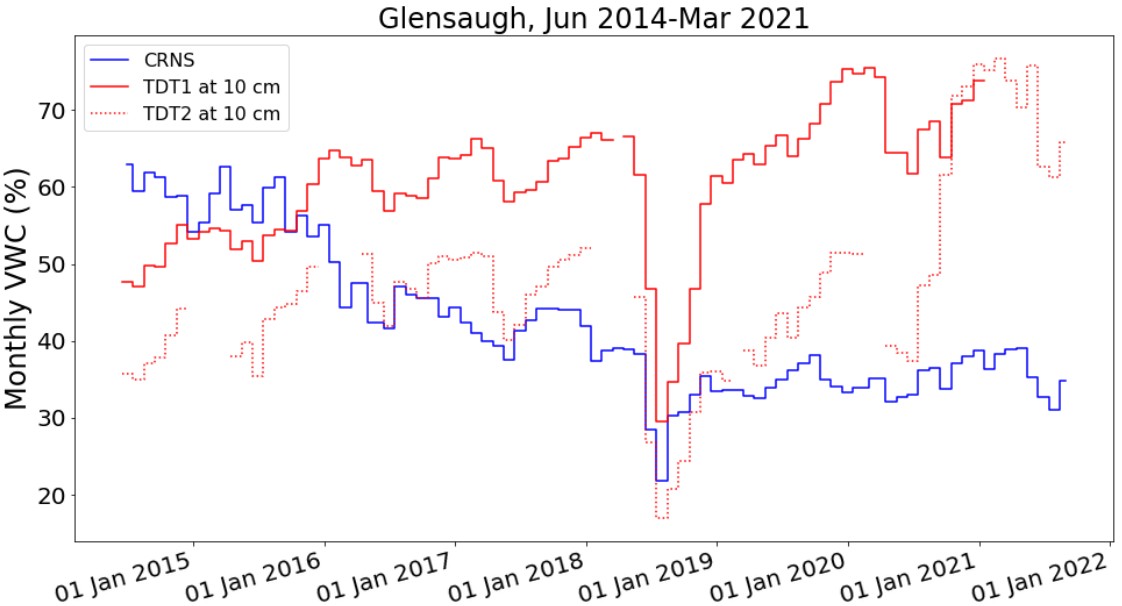

**Figure 4: VWC at Glensaugh SMS obtained using TDT soil moisture point sensors (red) and CRNS with *G* = 1 in Eq. (8) (blue).**

## 3 Results and discussion

Values of *G* have been derived at all 48 of the currently active COSMOS-UK sites. Reliable estimates of *G*, i.e. high t value
of estimate (estimate/standard error), cluster in range 1.0-1.5 (Fig. 5a). It is seen that the reliability of the estimates is highly
dependent on record length (Fig. 5b). In fact rather than record length, obtaining a reliable estimate of *G* requires there to have
been sufficient variation in *I* (Fig. 5c). Referring back to Fig. 1, this means that values of *G* derived for SMSs with records
starting from 2018 are not reliable, as since this time there has been little variation in the background count rate.

Where no reliable estimate can be obtained, alternatives are to use 1 or 1.2 (the mean of the reliable values). In COSMOS-UK,
the default value is 1.2 and this corresponds to the value of 1.19 derived by (Howat et al., 2018) based on 'regressions to the
global neutron monitor dataset'.

Investigations continue into the variations in derived *G* values. Preliminary studies show no simple relationship with
characteristics of the site such as latitude, longitude, altitude or cut-off rigidity.






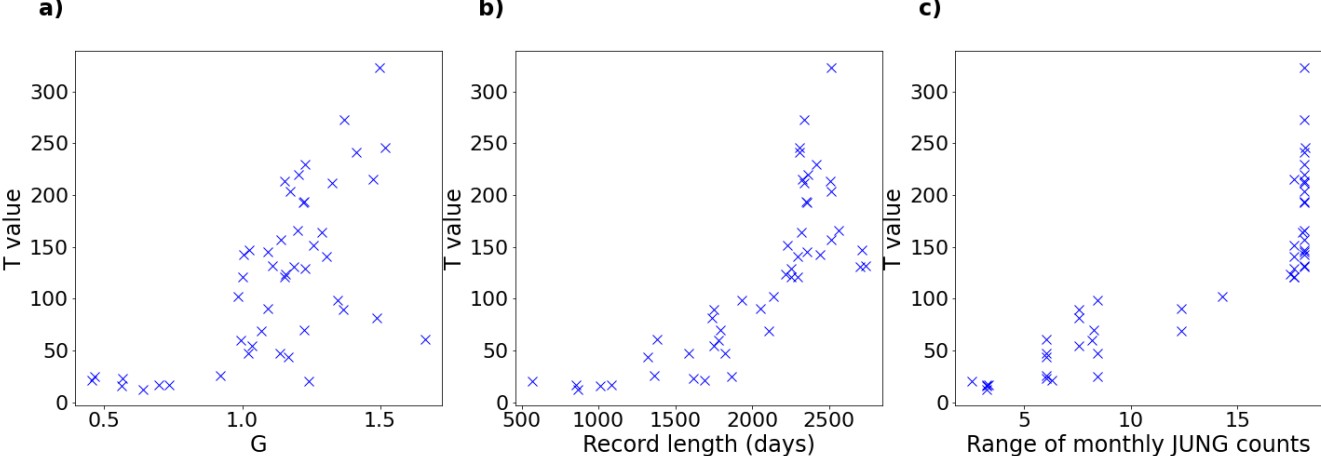

**Figure 5: T values of _G_ estimates for SMSs versus _G_ (a), record length (b) and range of monthly JUNG counts defined as a difference of maximum and minimum values of monthly counts (c).**

## 4 Conclusion

An empirical method has been developed to correct the incoming neutron flux used in deriving soil moisture from cosmic-ray sensors. A gradient, _G_, is estimated based on standardised count rates of both the RNMS and the SMS, and is therefore believed to better correct the neutron flux by recognising differences between RNMS and SMS characteristics. _G_ can be considered to be an estimate of the coefficient γ defined in other studies, but generally set to 1.

Applying this modification makes a minor change in derived VWC at sites with lower soil moisture, but at wetter sites the difference is more pronounced and in addition corrects an observed, but otherwise unexplained trend in the derived VWC time series. This modification will therefore further enhance the value of soil moisture derived using the CRNS method in many application areas.

There are numerous variations in the detail of the method used to derive _G_, for example how to normalise the count rates, what form of regression to use, and whether to sub-sample the data by season or wetness. While the methodology adopted here appears to work well in the temperate climate of the UK where soils are wet in most winters, other approaches may be more appropriate in other climates for example where soils regularly dry during a hot season.

Deriving _G_ from a more diverse set of sites may provide information on how _G_ varies between sites. It is hoped that other users of the CRNS method can undertake similar analyses to derive a larger data set that encompasses a wider range of site and sensor characteristics.



## Data availability

Daily and sub-daily COSMOS-UK hydrometeorological and soil data (2013-2019) can be downloaded from the EIDC at https://doi.org/10.5285/b5c190e4-e35d-40ea-8fbe-598da03a1185 (Stanley et al., 2021). For more recent data, reasonable

requests can be considered via https://cosmos.ceh.ac.uk/content/requesting-data.

Jungfraujoch neutron monitor data can be downloaded via https://www.nmdb.eu/nest/.

## Author contributions

JB and DB formulated the revised neutron intensity correction and carried out the initial investigation. MS helped further developing the method and wrote an analysis script in Python. JE contributed in discussions and understanding of the method.

DB wrote the first draft of the manuscript, while MS produced figures and finalised the manuscript with help from all co-authors.

## Competing interests

The authors declare that they have no conflict of interest.

## Acknowledgement

The authors gratefully acknowledge the contribution of the COSMOS-UK project team installing and maintaining the project infrastructure, and in the processing and archiving of the derived data.

The authors would like to thank the Physikalisches Institut, University of Bern, Switzerland for kindly and reliably providing Jungfraujoch neutron monitor data.

## Financial support

COSMOS-UK has been supported by the UK's Natural Environment Research Council (grant no. NE/R016429/1).

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
