# Peer review of "Technical note: A revised incoming neutron intensity correction factor for soil moisture monitoring using cosmic-ray neutron sensors"

_Hydrology and Earth System Sciences, 2021_

## Referee Comment (RC1)

Review of the manuscript hess-2021-564 A revised incoming neutron intensity correction factor for soil moisture monitoring using cosmic-ray neutron sensors by Szczykulska et al.,

General comments

The Authors present a revised incoming correction factor that could be used for improving soil moisture estimated with cosmic-ray neutron sensing (CRNS). The topic is interesting and timing and probably the study fits a technical manuscript in HESSD. The analysis is based on the extensive data-sets collected within the COSMOS-UK sites and the manuscript is generally also well structured and written.

Despite these general positive comments, I have some concerns regarding the scientific basis of this revised correction, its scientific impact and the further research activities that are discussed and suggested by the Authors. These major concerns are described below. I hope these comments will help clarify the scientific value of the present study and, eventually, to improve the overall quality of the manuscript.

Major concerns

[1] the study proposes a revised factor (eq.8) and the results are compared to the state-of-the-art (eq. 3). However, as far as I have understood, it is also well known an additional method (eq. 4) that so far it seems to be not widely adopted by the "CRNS community" but only used in some studies e.g., (Howat et al., 2018). So why do the Authors did not test this "not so common" revised method before proposing something new?

[2] as far as I have understood, the revised factor (eq.8) converges to what was currently available but not widely implemented (eq.4) not only in the form but, even more important, to the actual parameter i.e., G = 1.2 vs. gamma = 1.19 (See (Howat et al., 2018)). So, did I misunderstand or should the Authors be already satisfied by using eq. 4 without the need to propose a revised factor?

[3] for the development of the new revised factor (eq.8) the Authors compare incoming neutrons from RNMS (e.g., Jungfraujoch) to CRNS neutrons locally collected at a soil moisture site (SMS) during period where it is expected low variability due to soil moisture changes. Namely, removing local influences due to soil moisture, variability in the neutron counts should then be related to incoming fluctuations. The Authors then compare these local fluctuations to the RNMS. It is well discussed that, if these fluctuations are not the same, on a longer term, should be due to different cutoff rigidity and altitude between the RNMS and SMS. But since also eq.4 was developed to account for these factors, from my understanding it should be not a surprise that this revised method converge to eq. 4. So, overall, it seems to me that the Authors simply analyzed some time series and found empirically what is already know and addressed in literature. I might be wrong but, if this is the case, I encourage the Authors to clarify and improve the manuscript to better convey the novelty of the study.

[4] my last comment is related to the general assumption that incoming neutron counts from a RNMS adequately represents the relevant incoming neutron flux at the SMS and the revised factor accounts for some additional differences (L61-63). Based on that, the Authors conclude and suggest (L165-171) several research activities that could be performed for further improvements. Indeed I agree that using incoming fluctuation from RNMS is a first order correction that has to be considered also for CRNS applications. This assumption has however two shortcomings that should be considered. First, time series at RNMS need also several corrections that are still under investigations and the focus of current

research activities and improvements. Thus, these time series are not error-free. Second, some local incoming fluctuations at SMS are not detected by RNMS. Thus, these time series could not well inform local incoming fluctuations even in the case they were error-free. For these reasons, personally I do not see a good suggestion to push much effort in improving a method that is based on input (i.e., the time series at RNMS) that has these drawbacks. In contrast, I have seen that the use of alternative detectors installed directly at the SMS for the detection of incoming fluctuations has been suggested in literature using e.g., muons detectors (Stevanato et al., 2019; Stowell et al., 2021) or neutron spectrometers (Cirillo et al., 2021; Fersch et al., 2020). Personally, I believe that improving and working with these approaches could be much more valuable suggestions for further studies and developments instead of improving the manipulation of no error-free and non-representative time series from RNMS. Alternative, a discussion of the added value of the present revised factor in comparison to above-mentioned approaches should be reported.

References

Cirillo, A., Meucci, R., Caresana, Michele, Caresana, Marco, 2021. An innovative neutron spectrometer for soil moisture measurements. Eur. Phys. J. Plus 136, 985. https://doi.org/10.1140/epjp/s13360-021-01976-x

Fersch, B., Francke, T., Heistermann, M., Schrön, M., Döpper, V., Jakobi, J., Baroni, G., Blume, T., Bogena, H., Budach, C., Gränzig, T., Förster, M., Güntner, A., Hendricks Franssen, H.-J., Kasner, M., Köhli, M., Kleinschmit, B., Kunstmann, H., Patil, A., Rasche, D., Scheiffele, L., Schmidt, U., Szulc-Seyfried, S., Weimar, J., Zacharias, S., Zreda, M., Heber, B., Kiese, R., Mares, V., Mollenhauer, H., Völksch, I., Oswald, S., 2020. A dense network of cosmic-ray neutron sensors for soil moisture observation in a highly instrumented pre-Alpine headwater catchment in Germany. Earth System Science Data 12, 2289–2309. https://doi.org/10.5194/essd-12-2289-2020

Howat, I.M., de la Peña, S., Desilets, D., Womack, G., 2018. Autonomous ice sheet surface mass balance measurements from cosmic rays. The Cryosphere 12, 2099–2108. https://doi.org/10.5194/tc-12-2099-2018

Stevanato, L., Baroni, G., Cohen, Y., Cristiano Lino, F., Gatto, S., Lunardon, M., Marinello, F., Moretto, S., Morselli, L., 2019. A Novel Cosmic-Ray Neutron Sensor for Soil Moisture Estimation over Large Areas. Agriculture 9, 202. https://doi.org/10.3390/agriculture9090202

Stowell, P., Brown, A., Chadwick, P., Fargher, S., Rulten, C., Steer, C., Thompson, L.F., 2021. Low Cost Neutron and Muon Detectors for Soil Moisture Monitoring, in: Proceedings of 37th International Cosmic Ray Conference — PoS(ICRC2021). Presented at the 37th International Cosmic Ray Conference, SISSA Medialab, p. 1318. https://doi.org/10.22323/1.395.1318